# Integrin β4 as a Potential Diagnostic and Therapeutic Tumor Marker

**DOI:** 10.3390/biom11081197

**Published:** 2021-08-12

**Authors:** Haoyu Yang, Zixuan Xu, Yuqian Peng, Jiali Wang, Yang Xiang

**Affiliations:** 1School of Basic Medical Science, Central South University, Changsha 410013, China; 2303170110@csu.edu.cn (H.Y.); 2303170106@csu.edu.cn (Z.X.); pyuqian@csu.edu.cn (Y.P.); 2Xiang Ya School of Medicine, Central South University, Changsha 410013, China; wangjiali7@csu.edu.cn; 3Department of Physiology, School of Basic Medical Science, Central South University, Changsha 410013, China

**Keywords:** integrin β4, cancer, diagnostic significance, prognostic significance, therapeutic target

## Abstract

Integrin β4 (ITGβ4) is a class of transmembrane adhesion molecules composed of hemidesmosomes (HDs). Its unique long intracellular domain provides intricate signal transduction functions. These signal transduction effects are especially prominent in tumors. Many recent studies have shown that integrin β4 is differentially expressed in various tumors, and it plays a vital role in tumor invasion, proliferation, epithelial–mesenchymal transition, and angiogenesis. Therefore, we categorize the research related to integrin β4, starting from its structure and function in tumor tissues, and provide a basic description. Based on its structure and function, we believe that integrin β4 can be used as a tumor marker. In clinical practice, it is described as a diagnostic marker for the targeted treatment of cancer and will be helpful in the clinical diagnosis and treatment of tumors.

## 1. Introduction

The current cancer epidemiological data collected from the official databases of the World Health Organization and the American Cancer Society show that the overall risk of developing cancer for those aged 0–74 years is 20.2% (22.4% in men and 18.2% in women). A total of 18 million new cases were diagnosed in 2018, the most frequent of which were lung (2.09 million cases), breast (2.09 million cases), and prostate (1.28 million cases) cancer. In addition to sex-specific malignancies, the ratio of cancer frequency for men and women is greater than one for all cancers, except thyroid (i.e., 0.30). Regarding mortality, cancer is the second-highest cause of mortality worldwide (8.97 million deaths) after ischemic heart disease, but will likely be ranked first by 2060 (~18.63 million deaths) [1]. With the increases in morbidity and mortality, cancer is one of the main causes of human death and a major public health problem [2].

Integrins are heterodimeric receptors that consist of paired α and β subunits. In the human genome, 18α and 8β subunits are combined in a limited combination to provide 24 integrin receptors, each with its own specificity for selecting ECM or cellular adhesion proteins [3]. Integrin α6β4 plays a key role in the formation and stabilization of junctional adhesion complexes called hemidesmosomes (HDs), which are connected to the intermediate filament (IF) system [4]. This adhesion structure is critical to the integrity of the epithelial monolayer. In contrast to this function, many studies have shown that integrin α6β4 signals in various cancers to promote tumor invasion and metastasis [3]. Integrin β4 is closely related to lung cancer [5], breast cancer [6], prostate cancer [7], colon cancer [8], and other cancers. Differential expression of integrin β4 is found in these cancer tissues, and it is compatible with epidermal growth factor receptors in liposome interactions to promote cell growth and reproduction [6].

Integrin β4 has been widely reported as a prognostic and predictive molecule in patients with various cancers. The aim of this review was to evaluate the results from studies conducted on this subject. Using the search words “integrin β4”, “integrin β4 in tumor”, “diagnostic significance”, “prognostic significance”, and “therapeutic targets”, a search was performed on PubMed. Finally, 105 articles were found. This review regards integrin β4 as a potential tumor marker, providing assistance with the diagnosis and treatment of tumors in the future, and helping to solve major public health problems in the world today.

## 2. Overview of Integrin β4

### 2.1. Structure of Integrin β4

A total of 24 different kinds of integrins have been described [9]. The cytoplasmic tail of human integrin subunits is less than 75 amino acids in length, but the β4 tail is an exception, with a length of approximately 1000 amino acids, which includes four fibronectin type III repeats [10]. It contains a large extracellular domain, a transmembrane area, and a cytoplasmic area. The extracellular domain binds to extracellular matrix (ECM) proteins, such as laminin; the cytoplasmic area interacts with kinases, such as cytoplasmic tyrosine kinase. This special structure, as well as the diversity of the internal and external structures of the connected cells, results in integrin α6β4 having a unique cytoskeleton and signal transduction function [9].

### 2.2. Function of Integrin β4

Integrins are a superfamily of cell adhesion receptors. They mainly recognize extracellular matrix ligands and cell surface ligands. They mainly act as traction receptors and can transmit and detect changes in the mechanical forces acting on the extracellular matrix [10].

Integrin β4 only binds to integrin α6 to form a heterodimer [9], and its extracellular domain usually binds to the basement membrane component laminin-332, which is the main HD component [11]. Laminin interacts with integrin α6β4 to maintain the integrity of epithelial cells, affects cell adhesion functioning, and regulates cell proliferation [12]. Integrin β4 is a transmembrane protein and plays the role of signal transduction. Therefore, integrin β4 has the ability to promote HD composition and stabilize epidermal adhesion [9].

Through the intracytoplasmic region of integrin β4, it interacts with ECM molecules to activate many intracellular signal mediators, including FAK, src, PKC, PI3K, and MAPKs, resulting in alterations in the actin cytoskeleton and gene expression [12]. For example, critical for the mechanical stability of HDs is the interaction between integrin α6β4 and plectin, which is destabilized when HD disassembly is required. Growth factors such as epidermal growth factor (EGF) can trigger HD disassembly and induce the phosphorylation of integrin β4 intracellular domains, which mediate invasion in carcinoma cells [13].

Therefore, integrin β4 can activate intracellular signal transduction in normal tissues and can reorganize the actin cell signal skeleton to maintain the integrity of epithelial cells. Recent studies found that integrin β4 promotes cell proliferation, migration, and invasion and plays a pivotal role in tumorigenesis, leading to poor prognosis and reduced survival rates [14].

## 3. Differential Expression of Integrin β4 in Tumors

Integrin β4 is overexpressed in a variety of human cancers. In many cases documented in studies, it is positively correlated with poor prognosis [3].

### 3.1. Lung Cancer

Both quantitative mass spectrometry analysis and Western blot analysis showed that integrin β4 is abundantly present in lung cancer cells. Clinical information was available on 21 patients with a median age at diagnosis of 63 years (range: 44–79 years). Survival information was available for 20 patients. The five-year overall survival for MUC5AC-negative patients was 93% (95% confidence interval = 59–99%), compared to 67% in the MUC5AC-expressing patients (95% confidence interval = 19–90%). MUC5AC is a class of mucins that are secreted and polymerized to form gels in the airways. It is regulated at the transcriptional, posttranscriptional, and epigenetic levels, and posttranslational modifications play an important role in mucin binding and the clearance of microbes and pollutants [15]. It is overexpressed in lung cancer cells, interacts with integrin β4 to initiate FAK phosphorylation, and mediates the invasion and migration of lung cancer cells through chitosan separation of tumor stem-like cells [5,16]. TP53 mutation and integrin β4 overexpression co-occur in many aggressive malignancies, and lung squamous cell carcinoma has a higher TP53 mutation frequency; the overexpression of integrin β4 can also lead to venous infiltration and a reduction in overall survival in patients with non-small cell lung cancer [17].

### 3.2. Breast Cancer

The role of integrin β4 in breast cancer progression has been well established [18]. Studies showed that integrin β4 participates in the growth and branching of the breast and plays a key role in the development of the breast [19]. In transitional breast cancer cells, integrin β4 is expressed, but not in non-invasive breast cancer cells or normal breast epithelial cells [20]. The tissue of triple-negative breast cancer (TNBC) is also rich in integrin β4 [21].

Integrin β4 is used to identify more aggressive subtypes of mesenchymal cancer cells [21]. Sung et al. found that in a co-transplant mouse model, human breast cancer cell line MDA-MB-231 with cancer-associated fibroblasts (CAF) produced a larger tumor mass than ITGB4 knockdown MDA-MB-231 [22].

Integrin β4 can induce an invasive status of breast cancer cells. The acyltransferase DHHC3 transfers palmitate as palmitoyl-CoA to the cysteines of integrin β4, which induces its palmitoylation and then triggers the invasive status of breast cancer cells [23]. Cell viability and motility are enhanced when estrogen induces integrin β4 to promote the phosphorylation of AKT [24]. Integrin β4 physically interacts with the ecto domain of nectin-4, and then promotes angiogenesis in metastatic breast cancer stem cells (mBCSCs) via the Src, PI3K, AKT, and iNOS pathways [25].

### 3.3. Prostate Cancer

Integrin β4 also plays a vital role in the occurrence and development of prostate cancer. Studies found that integrin β4 promotes prostate tumorigenesis by co-expressing with ErbB2 and c-Met in tumor progenitor cells [26]. A compelling body of evidence indicates that integrin β4 is a key player in prostate tumor development. Some data show a large CpG island present in the integrin β4 promoter in prostate tumor development, which mediates the differential transcription of the integrin β4 gene [27]. A study found that integrin β4 knockdown attenuated both cell migration and invasion but did not affect proliferation in castration-resistant prostate cancer cells [28]. All these results indicate that integrin β4 is closely related to the occurrence and development of prostate cancer.

### 3.4. Colon Cancer

Various forms of integrin α6β4 can differentially regulate intestinal epithelial cell functions under both normal and pathological conditions [24]. A study showed that two forms of integrin β4 exist in the human intestinal epithelium: a full-length 205 kDa β4A subunit expressed in differentiated enterocytes, and a novel β4A subunit that does not contain the COOH- terminal segment of the cytoplasmic domain (integrin β4A (ctd-)), which is found in undifferentiated crypt cells [8]. The integrin β4 ctd- form was identified in normal proliferative colonic cells but was found to be predominantly absent in colon cancer cells, whereas the level of the wild-type form of integrin β4, which is required for adhesion to laminin, is increased in primary tumors [29]. In addition, focal adhesion kinase (FAK) initiated by integrin β4 was found to be expressed in the nuclei of colon cancer cells, indicating that the activation of integrin β4 downstream molecules is also associated with abnormal cell proliferation [30], and other pathways such as miRNA-21-integrin β4-PDCD4 can affect cell viability [31]. When integrin β4 decreased sialylation by NEU1, it was accompanied by decreased phosphorylation of integrin followed by the attenuation of focal adhesion kinase and the Erk1/2 pathway, which led to the suppression of metastasis [32].

The expression and function of integrin β4 in the tumors listed above are described in Table 1.

## 4. Integrin-β4-Mediated Cancer Progression

To explore the downstream signaling pathways of integrin β4, cBioPoral was used to detect some genes related to integrin β4. Among them, integrin β4 is strongly correlated with EGFR, which amplifies proliferative and invasive signaling. EGFR-family genes EGF, EGFR, and ErbB3 are also strongly related to integrin β4. In the work map, a number of laminin subunits (LAMA3, LAMB3, and LAMC2) connect with integrin β4. This is consistent with the finding that integrin β4 binds laminin-332 in the extracellular matrix, which indicates that integrin β4 plays a vital role in the stabilization of cells, which means that cell invasion and migration can be limited [17]. Phosphoinositide 3-kinase (PI3K) activity is stimulated by diverse oncogenes and growth factor receptors, and elevated PI3K signaling is considered a hallmark of cancer [33]; its pathway genes PIK3CA and AKT1, and other known signal partners, such as MET and FYN, etc., were found to be strongly related to integrin β4 [17].

### 4.1. Integrin β4 Is Associated with Tumor Invasion and Migration

Overexpression of integrin α6β4 is related to aggressive tumor behavior and poor prognosis [34]. Six hours after a single layer of epithelium was scratched, a burst of integrin β4 expression was observed, especially in cells at the edge of the wound, accompanied by an increase in chemical kinetic migration [35].

In the process of cancer progression, integrin α6β4 is released from hemidesmosomes, responds to the wounded epithelium or cancer progression, and is located in the F-actin protrusions [36,37,38,39,40]. It binds to the actin cytoskeleton, where it activates RhoA, mediating membrane ruffling, lamellae formation, and generating traction, thereby causing cell invasion and metastasis [41]. Various interactions between the extracellular domain of integrin β4 and the matrix play an important role in tumor invasion and migration.

Integrin β4 can promote the invasiveness of certain malignant tumors, such as squamous cell carcinoma, breast cancer, and gastric cancer, through the MAPK and NF-κB activation pathways [40,42,43,44]. Li et al. used bioinformatics analysis and found approximately 70 pathways significantly dysregulated when integrin β4 expression was high. The MAPK pathway and propanoate metabolism were found to be located in the network center of these pathways [45]. Integrin β4 does not trigger any pathway, but it can bind to different proteins, including human leukocyte antigen-1(HLA-1) [46], epidermal growth factor receptor EGFR [13], extracellular matrix protein ECM1 [47], etc. Human leukocyte antigen-1 (HLA-1) binds to the integrin β4 subunit and then mediates the activation downstream. When its extracellular domain is stimulated, phosphorylation of Src kinase is induced and focal adhesion kinase (FAK) starts, then activating PI3K/AKt and Erk. These enzymes can mediate the occurrence of tumor cell migration [46]. Integrin β4 EGFR/Src signaling mediates the tyrosine phosphorylation of integrin β4, and then recruits FAK to integrin β4, leading to FAK activation and signal transduction, thereby activating the AKT signaling pathway and promoting tumor invasion [13,48,49]. The integrin β4/FAK/SOX2/HIF-1α signaling pathway also regulates the metastasis of tumor cells [47]; integrin β4 overexpression triggers increases in the expression of matrix metalloproteinase-2 (MMP-2) and MMP-9 through the ERK1/2 pathway, the degradation of the extracellular matrix, and the invasion ability of tumor cells [50,51] (Figure 1).

The extracellular domain of integrin β4 has potential glycosylation sites. A report showed that N-glycan deletion on integrin β4 impaired integrin-β4-dependent cancer cell migration, invasion, and growth in vitro and diminished tumorigenesis in vivo, which means that N-glycosylation is important in cancer progression. It activates the intracellular PI3K pathway to promote cell invasion and migration [14] and Ca^2+^/PKC; the Sp1/CaM and Akt/GSK-3β/Snail1 signaling pathways are also activated when glucosamine triggers integrin β4/plectin complex reduction [52].

Certain endocrine substances can also affect tumor cell invasion. For example, the N-terminal truncated subtype of the p63 transcription factor ΔNp63 acts as a transcription factor for integrin β4, and estrogen can activate estrogen receptor α (ERα) transcription, thereby inducing ΔNp63 expression; then, ΔNp63 induces the expression of integrin β4, which leads to the phosphorylation of AKT and enhances cell viability and motility [24]. Changes in integrin β4 in cancer can also mediate the inhibition of miR-29a through a TOR-dependent mechanism, thereby mediating the increase in its downstream signaling molecule SPARC, which is a secreted glycoprotein that results in tumor cells becoming aggressive phenotypes [53].

The invasive and metastatic phenotype promoted by integrin α6β4 signaling is mediated by the phosphorylation of the cytoplasmic tail of the integrin β4 subunit [3]. As mentioned above, integrin β4 is involved in the formation of HDs. HDs promote the stable adhesion of basal epithelial cells to basement membranes (BMs). The key to the mechanical stability of HDs is the interaction between integrin α6β4 and plectin. Integrin α6β4 binds to skeletal connexin and to keratin filaments, stabilizing the adhesion of cells to the epidermal basement membrane [54]. When damage occurs, causing the HD structure to disassemble, the interaction becomes unstable. Growth factors such as epidermal growth factor (EGF) can trigger HD disassembly and induce the phosphorylation of integrin β4 intracellular domains; serine phosphorylation seems to be the main mechanism regulating HD instability [13,49]. Usually, tyrosine kinase mediates the phosphorylation process [46,55]. Disruption of the α6β4–plectin interaction through phosphorylation of the β4 subunit results in a reduction in the adhesive strength of keratinocytes to laminin-332 and the dissolution of HDs, resulting in enhanced cell motility [54].

### 4.2. Integrin β4 Is Associated with Tumor Cell Proliferation

A study showed that the proliferation of tumor cells is promoted at a relatively low concentration when integrin β4 interacts with netrins, which are secretory molecules of laminin [56]. For example, in glioblastoma, netrin-4 expression was downregulated, thus reducing its inhibitory effect on cell proliferation. In parallel, the expression of integrin β4 is upregulated, which sensitizes the cells to low concentrations of netrin-4 for maintaining cell proliferation [57]. Triple-negative breast cancer cells overexpressing integrin β4 provide integrin β4 to cancer-related fibroblasts (CAF) through exosomes, which induces the phosphorylation of c-JUN or AMPK in CAF. As a result, the proliferation of breast cancer cells is enhanced [22].

### 4.3. Integrin β4 Is Related to Epithelial–Mesenchymal Transition

Epithelial–mesenchymal transition (EMT) is an important change in cell phenotype. It frees epithelial cells from the structural constraints imposed by the tissue structure. Its core elements include the dislocation of desmosomes and a reduction in intercellular adhesion. More importantly, after EMT, cells show an increase in motor potential. Therefore, EMT is a fundamental process that underlies development and cancer [58]. Studies showed that EMT is closely related to the invasion and metastasis of pancreatic cancer, colon cancer, breast cancer, and other tumors [59,60].

TGF-β1 is an important molecule in the initiation of EMT [61]. It is now well accepted that the loss of E-cadherin, EMT marker α-SMA, and connective tissue growth factor (CTGF) expression are key events in the EMT process. Moreover, it was reported that by stimulating human renal proximal tubular epithelial (HK-2) cells with TGF-β1, the expression of E-cadherin was downregulated and the expression of α-SMA and CTGF were upregulated in a dose-dependent manner. Additionally, stimulating TGF-β1 can increase the expression of integrin β4 [62]. It can control the expression of the integrin β4 subunit and can activate ErbB2- and FAK-dependent clusters through a pathway triggered by epithelial growth factor-dependent phosphorylation, thereby inducing crosstalk between integrin β4 and growth factor receptors [61,62,63,64,65,66]. Takaoka revealed that the expression of integrin β4 is closely related to the methylation of β4 promoter [67], which contains CpG islands, and it can be regulated dynamically during the EMT of mammary gland cells triggered by TGF-β [68].

Moreover, enhanced ECM1 expression facilitates gene expression levels associated with EMT. ECM1 directly interacts with integrin β4 and activates the ITGB4/FAK/SOX2/HIF-1α signal pathway mentioned above [47].

Additionally, a zinc-finger transcription factor, containing KRAB and SCAN domains, ZKSCAN3, directly binds to the integrin β4 promoter and enhances its expression; then, integrin β4 triggers FAK to activate the AKT signaling pathway to promote EMT progression in hepatocellular carcinoma [69].

### 4.4. Integrin β4 Is Associated with Angiogenesis

Tumor angiogenesis is an important condition for tumorigenesis and is crucial for tumor growth [70,71]. Integrin β4 can induce angiogenesis through two different pathways: NF-κB and phosphorylated Erk nuclear translocation, and the Src, PI3K, AKT, and iNOs pathways [72]. For example, if nectin-4 is expressed on the surface of metastatic breast cancer cells, its extracellular domain falls off and is released into the microenvironment. It combines with integrin β4 expressed on the surface of human umbilical vein endothelial cells to activate Src, PI3K, AKT, and iNOs. This induces angiogenesis [25].

Dysfunction of the vascular endothelial cell (EC) barrier is regarded as a key feature of tumor angiogenesis, and lipid rafts receptor c-Met is reported to mediate increases in EC barrier integrity. According to Yulia Ephstein et al., integrin β4 is an essential participant in this process [73]. There are other studies showing that integrin α6β4 induces endothelial cell migration and invasion by promoting the nuclear translocation of P-ERK and NF-κB, which promotes the branching of medium- and small-sized vessels from β4+ to β4- microvessels. This indicates that integrin α6β4 signaling promotes the onset of the invasive phase of pathological angiogenesis [72].

## 5. Potential Clinical Applications of Integrin β4

From the above description of the relationship between integrin β4 and tumors, it is known that integrin β4 is differentially expressed in many tumors. Its extracellular domain constitutes the cytoskeleton and affects the matrix structure. The intracellular domain activates various signaling pathways to produce different cell behaviors. Integrin β4 has an inseparable relationship with tumors. Therefore, integrin β4 is regarded as a potential tumor marker and plays a vital role in the diagnosis and treatment of tumors.

### 5.1. Diagnosis

Integrin β4 can regulate key signaling pathways related to cancer progression and promote the migration, invasion, and proliferation of cancer cells [4,42]. Studies showed that the tissues of TNBC patients are rich in integrin β4, and integrin-β4-related gene classification was proven to be useful for analyzing the survival and prognosis of breast cancer patients. It can also be used as a separate cell surface marker to distinguish different subtypes of TNBC cells [6]. TP53 mutation and integrin α6β4 overexpression occur concurrently in many aggressive malignancies. The frequency of TP53 mutation in lung squamous cell carcinoma (SCC) is very high. Other researchers studied the relationship of integrin β4 expression with clinicopathological characteristics and the survival of non-small cell lung cancer (NSCLC), finally proving that integrin β4 is overexpressed in NSCLC and is a marker of poor prognosis [17].

In malignant tumors, lymphatic vessel infiltration (LVI) and peripheral nerve infiltration (PNI) are two important pathological parameters that require accurate assessment. Studies found that both LVI and PNI are clearly and accurately outlined by integrin β4 immunostaining. Integrin β4 is expressed in most tumor cells, which can identify LVI and PNI that are invaded by small tumor clusters. The applicability of integrin β4 was also confirmed in nine other types of malignant tumors, including colon cancer, prostate cancer, esophageal cancer, lung cancer, renal cancer, uterine cancer, tongue cancer, bladder cancer, and liver cancer. Integrin β4 has therefore become a reliable marker for the simultaneous detection and diagnosis of LVI and PNI [74].

As the upstream transcription factor of integrin β4, miR-21 significantly affects the expression of integrin β4, thereby affecting the migration of colon cancer cells. The tumor suppressor pdcd4 inhibits transformation and invasion and is downregulated in cancer. It is a newly identified tumor suppressor gene involved in the occurrence of colon cancer [75]. miR-21 exists in the 3′ non-coding segment. In colorectal cancer cell lines, miR-21 is negatively correlated with the protein Pdcd4 [76]. Therefore, except for indicating cancer progression by integrin β4, the combination of high miR-21 and low ITGβ4 and PDCD4 expression can predict the presence of metastasis and can be used as a predictive tool for colorectal cancer metastasis [77].

The aforementioned mucin MUC5AC, which interacts with integrin β4, was also found to affect the expression of focal adhesion kinase (FAK) in the downstream signaling pathway of integrin β4, leading to the migration of lung cancer cells. Therefore, mucin MUC5AC co-localized with integrin β4 is a marker of poor prognosis in lung cancer [16].

Integrin β4 participates in multiple signaling pathways and can also enhance them, including ErbB2 [78], PI3K [79], FAK/AKT [48], and c-Met [73], to promote tumor progression. It is a marker of poor prognosis in pancreatic ductal carcinoma [80] and breast cancer [81]. Therefore, detection of the expression of integrin β4 can also be used in the diagnosis of a variety of tumors.

### 5.2. Treatment Targeting Integrin β4

Integrin β4 is differentially expressed in tumors and significantly affects the prognosis and survival rate of patients. For example, Kaplan–Meier curves and univariate analysis demonstrated that high ITGB4 expression was significantly associated with unfavorable overall survival in colon cancer (HR = 1.292, 95% CI = 1.084–1.540, *p* = 0.004) [82]. Integrin β4 promotes tumor growth and metastasis, suggesting that integrin β4 may be a prognostic marker or treatment target for patients with carcinoma [83]. Immunohistochemical analysis of 134 cases of surgically resected pancreatic ductal adenocarcinoma (PDA) showed that integrin β4 expression in tumors is heterogeneous; such tumors often show strong diffuse integrin β4 expression [80]. Other researchers identified 43 differentially expressed cancer-related genes in adenocarcinoma and squamous cell carcinoma, and upregulation of integrin β4 was observed [84]. This difference in the expression of integrin in cancer patients and its impact on prognosis indicate its potential as a new target and prognostic marker for the treatment of cancer patients [83].

#### 5.2.1. Potential Targets for Integrin β4 Pathway Therapy

ARRDC3 is a metastasis inhibitor that can induce NEDD4-dependent ubiquitination of ITG β4 and targeting of endosomal integrin β4 into lysosomes, thereby reducing the level of integrin β4 in extracellular vesicles, to reduce the metastasis potential of the extracellular vesicles of breast cancer cells. ARRDC3 can also reduce the metastatic potential of breast cancer cell-derived integrin β4 + EVs, which are mainly detectable from supernatants of TNBC by reducing integrin β4 levels in EVs. Therefore, the ARRDC3/ITG β4 pathway has shown potential for the targeted therapy of TNBC [18]. FAK is a non-receptor tyrosine kinase that acts as a convergence point for various signal transduction pathways related to cell adhesion, migration, and oncogenic transformation [85,86,87]; its overexpression is closely related to the malignancy degree of TNBC [44,88]. Studies showed that the integrin β4/FAK complex regulated by EGF/Src signaling is closely related to TNBC and is a new target for future treatment. Integrin-β4-dependent mitochondria are a potential location of cancer metabolism [89]: integrin β4 triggers BNIP3L-dependent mitochondrial phagocytosis. These mitochondria trigger the aerobic glycolysis of TNBC in CAF and initiate tumor metabolism. Therefore, mitochondria induced by integrin β4 can also be used as targets for cancer treatment [22].

In the serum and tissues of gastric cancer patients, the level of extracellular matrix protein (ECM1) is positively correlated with tumor invasion and recurrence. It directly interacts with integrin β4 to activate the integrin β4/FAK/glycogen synthase kinase 3β signaling pathway and to induce the expression of transcription factor SOX2, thereby enhancing the activity of the hypoxia-inducible factor alpha (HIF-1α) promoter to regulate the metastasis of gastric cancer cells. This is significant for the development of therapeutic targets to prevent tumor metastasis and recurrence [47]. Ribosomal protein S27 (MPS-1) is highly expressed in gastric cancer cells and is related to the metastasis of gastric cancer cells. Integrin β4 is the downstream target of MPS-1 and mediates its metastatic effect on cells. The MPS-1/integrin β4 signal axis may also become a therapeutic target for gastric cancer [90].

Integrin β4 and its phosphorylated form can promote cell migration and invasion in pancreatic cancer. p-integrin β4-Y1510 integrin β4 phosphorylated at tyrosine position 1510) can regulate the downstream MEK1-ERK1/2 signaling cascade and affect tumor blood vessel generation, invasion, and metastasis [63,91,92,93]. Therefore, targeting integrin β4 or its phosphorylation on Y1510 may also become a new treatment option for pancreatic cancer [94].

SEC is a small chiral molecule. By inducing integrin β4 nuclear translocation, nuclear integrin β4 can bind to the ATF3 promoter region to activate ATF3 expression, upregulate downstream pro-apoptotic genes, and promote the apoptosis of cancer cells expressing high levels of integrin β4. It also promotes the binding of annexin A7 (ANXA7) to integrin β4, activates ANXA7, and triggers the phosphorylation of integrin β4 to promote nuclear translocation. Therefore, SEC is a potential target drug to treat integrin-β4-related tumors [95].

Studies showed that integrin β4 is a transcription target of TAp73 [96]. TAp73 is an important member of the p53 family and plays a key role in tumor suppression [97]. It is a tumor suppressor transcription factor that induces apoptosis and cell cycle arrest. Its mutations rarely occur in human cancers; conversely, it is upregulated in many cancers [98]. TAp73 can also be used as a potential target for the treatment of integrin-β4-related tumors.

Based on the numerous pathways mediated by integrin β4, blocking the signal pathway is also a reasonable approach to treating cancer. For example, integrin β4 combines with a variety of tyrosine kinases (including ErbB2, EGFR, and Met) to enhance its signaling function. Genetic studies showed that integrin β4 signaling promotes both angiogenesis and tumorigenesis. Therefore, some researchers proposed that blocking integrin β4 and receptor tyrosine kinase signaling simultaneously is a potential rational therapy for cancer [99]. NECTIN-4 is encoded by the pvr1-4 gene. Under hypoxia, the expression of ADAM-17 driven by metastatic breast cancer stem cells (mBCSCs) leads to the shedding of the extracellular domain of nectin-4 into the microenvironment and interacts with integrin β4 in the microenvironment. It interacts on endothelial cells and promotes angiogenesis through the Src, PI3K, AKT, and iNOS pathways. As such, disrupting the interaction between the extracellular domain of nectin-4 and integrin β4 is a means to target mBCSCs to resist angiogenesis [25].

#### 5.2.2. Drug Intervention of the Integrin β4 Pathway

Integrin β4 interacts with certain drugs and may be a molecule that induces tumor resistance. Gefitinib is one of the representative drugs of EGF receptor inhibitors (EGFRs), which is a transmembrane protein including an extrinsic ligand-binding domain, a single hydrophobic transmembrane region, and an intrinsic tyrosine kinase domain. It is usually used as an antiproliferative agent and induces cell apoptosis. It is used to treat patients suffering from cancer due to EGFR mutations. However, in clinical treatment with gefitinib, drug resistance appears in most patients [100]. Studies showed that integrin β4 interacts with EGFRs: both of them have a large cytoplasm domain with several phosphorylation sites at the serine and tyrosine residues. According to Jia Huafeng and Zhang Deqing, integrin β4 is related to the sensitivity of gefitinib to gastric cancer; they found an obvious negative correlation between p-EGFR and integrin β4, which illustrates the direct interaction between p-EGFR and integrin β4, which contributes to the proliferation and growth of cancers, because gefitinib, a single molecular targeted drug that inhibits EGFRs, is unable to resist the proliferation and growth of tumor cells through upregulating the expression of integrin β4 [100]. Integrin β4 interacts with EGFRs in the form of independent ligands and is overexpressed and induces drug resistance in hepatocellular carcinoma [48].

Berberine is an extract with a long history of use in Chinese herbal medicine. In the past few years, berberine was shown to have therapeutic effects on many types of cancers, such as osteosarcoma, prostate cancer, liver cancer, etc. [101,102,103]. Studies showed that berberine induces the expression of integrin β4 and PDCD4 in colon cancer cell lines by inducing cell apoptosis and caspase-3 activity, and it inhibits the activity of miR-21 in colon cancer cells. As described above, high miR-21 and low ITGβ4 and PDCD4 expression mark the metastasis of cancer. Therefore, berberine inhibits the activity of colon cancer cells [31].

Integrin β4 palmitoylation is a post-translational modification required for lipid raft positioning and signaling activity, which can enhance the invasive potential of cancer cells [104,105]. Integrin β4 is the target of curcumin in aggressive breast cancer cells [104]. Curcumin can block integrin β4 palmitoylation in invasive breast cancer cells and block multiple phosphorylation signaling cascades [106], but the occurrence of palmitoylation does not rely on any phosphorylation or stimulation of growth factors. It is not affected by upstream signal conduction. Curcumin has therefore become an inhibitor of breast cancer cell invasion and metastasis [23].

#### 5.2.3. Physical Intervention in the Integrin β4 Pathway

Studies showed that after ionizing radiation exposure, cell integrin β4 tyrosine residue 1510 is phosphorylated, which leads to the activation of the integrin α6β4–Src–AKT signaling pathway, and then leads to the disappearance of p21 targeted by p53, prevents the activation of photoproteinase, and causes tumor cells to age during radiation exposure. Therefore, integrin β4 and PI3K/AKT may be important targets for radiotherapy [107].

## 6. Conclusions

In the past few years, the prominent differential expression of integrin β4 in various tumors has attracted the attention of researchers. Therefore, research on integrin β4 has concentrated on the field of tumors. Scientists have discovered the differential expression of integrin β4 and its related upstream and downstream molecular pathways in different types of cancer models. As far as the current research status is concerned, the role of integrin β4 in various tumors is relatively clear. Integrin β4 regulates various tumorigenesis processes, such as invasion, migration, proliferation, epithelial–mesenchymal transition, and tumor angiogenesis. With its widespread presence in tumors, we suggest that integrin β4 can be used as a non-specific tumor marker for the diagnosis and treatment of tumors.

## Figures and Tables

**Figure 1 biomolecules-11-01197-f001:**
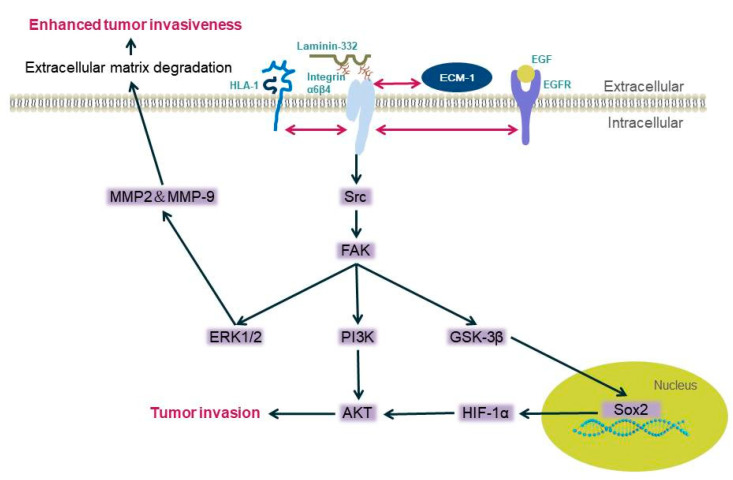
Diagram of the mechanism of integrin β4 mediating tumor cell invasion. Integrin α6β4 binds with laminin-332, and when the pathways are activated, the binding between laminin and α6β4 is not stable enough, and cells are more likely to be invasive (the double-headed arrow in the figure represents the interaction between the two proteins).

**Table 1 biomolecules-11-01197-t001:** Expression level and function of integrin β4 in different cancers.

	Lung Cancer	Breast Cancer	Prostate Cancer	Colon Cancer
Expression Level	① Abundantly present ② Strongly overexpressed	① Occurs in transitional breast cancer cells ② Not in non-invasive breast cancer cells	Overexpressed (has a large CpG island)	① Wild-type form is increased ② Integrin β4A(ctd-) is predominantly absent
Function	① Initiates phosphorylation, invasion, and migration ② Occurs with gene mutation	① Induces invasive status ② Enhances cell viability and motility ③ Promotes angiogenesis	① Induces tumorigenesis ② Induces cell migration and invasion ③ Does not affect cell proliferation	① Induces abnormal cell proliferation ② Affects the cell viability ③ Induces metastasis

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
