# Peer review of "Integrin β4 as a Potential Diagnostic and Therapeutic Tumor Marker"

_biomolecules, 2021, doi:10.3390/biom11081197_

Round 1
Reviewer 1 Report
The manuscript by Yang et al, entitled “Integrin b4 as a potential diagnosis and therapeutic tumor marker” gathers the results of the recent studies on the role of a6b4 integrin in cancer. This review is timely and provides a relevant perspective that should be of interest to the field. However, several parts of this manuscript are confusing because there are incorrect descriptions and insufficient explanations.
Comments
- Please correct the spelling of the word “therapeutic” in the title.
- Integrinb4 should be integrin b4 through the manuscript.
- Line 99, “or mediates the invasion and migration……like cells” Is this correct?
- Line 113, “In the present …….MDA-MB231.” please describe the contents in ref 21 more precisely using the term, CAF for avoiding readers’ misreading .
- Line 145, what is meant by “the stabilization of cells”? Please either rewrite this phrase or give a brief explanation.
- Line 148, write gene names in capitals.
- Figure 1 shows that HLA-1 but not integrina6b4 seems to activate ERK1/2, AKT Src signals. Is this correct?
- Line 190-197, this paragraph is unnecessary, as it is not directly relevant to the topic of this paper, cancer.
- Line 217, “Usually tyrosine kinase…..phosphorylation process” is difficult to understand and should be rewritten.
- Figure 2, laminins associate with integrins through G-domain but not coiled-coil domain. Please modify it.
- Line 259, please add more references about the relationship between EMT and integrin b4 because most of the references are about other integrins.
- Line 283, the word “Therefore” is unclear. What is the relationship between TP53 and NSCLC?
- Line 323, ref 76 describes PDA while ref 77 describes lung cancer. These topics should be described in a separate paragraph.
Author Response
Response to Reviewer 1 Comments
Point 1:Please correct the spelling of the word “therapeutic” in the title.
Response 1: We have corrected it
Point 2:Integrinb4 should be integrin b4 through the manuscript.
Response 2: We have corrected it
Point 3:Line 99, “or mediates the invasion and migration……like cells” Is this correct?
Response 3: After researching for other articles, we’ve changed “or” into “and” because we found that the migration and invasion of cancer cells happens after the phosphorylation of FAK
Point 4:Line 113, “In the present …….MDA-MB231.” please describe the contents in ref 21 more precisely using the term, CAF for avoiding readers’ misreading .
Response 4: We have corrected it.
Point 5:Line 145, what is meant by “the stabilization of cells”? Please either rewrite this phrase or give a brief explanation.
Response 5: We have changed it into “maintaining cell position”
Point 6:Line 148, write gene names in capitals.
Response 6: We have corrected it
Point 7:Figure 1 shows that HLA-1 but not integrina6b4 seems to activate ERK1/2, AKT Src signals. Is this correct?
Response 7: After researching for many articles, we seriously re-combine the pathway in the figure. Integrin β4 can interact with many proteins expressed outside the cell membrane, these interactions could trigger different pathways, and our figure shows some of the common ones.
Point 8:Line 190-197, this paragraph is unnecessary, as it is not directly relevant to the topic of this paper, cancer.
Response 8: We have corrected it.
Point 9:Line 217, “Usually tyrosine kinase…..phosphorylation process” is difficult to understand and should be rewritten.
Response 9: We have deleted the former “phosphorylation” because we typed an extra “phosphorylation” in this sentence.
Point 10:Figure 2, laminins associate with integrins through G-domain but not coiled-coil domain. Please modify it.
Response 10: We regretfully deleted this picture because of some copyright issues
Point 11:Line 259, please add more references about the relationship between EMT and integrin b4 because most of the references are about other integrins.
Response 11: We have done our utmost to find some other references about EMT and revised it.
Point 12:Line 283, the word “Therefore” is unclear. What is the relationship between TP53 and NSCLC?
Response 12: We have considered it seriously and changed the sentence into “Other researchers...”. What we wrote this way is to show that many pathways mediated by integrin β4 are differentially expressed in cancer cells, thus showing that it can be used as a diagnostic target
Point 13:Line 323, ref 76 describes PDA while ref 77 describes lung cancer. These topics should be described in a separate paragraph.
Response 13: We have modified this paragraph
Reviewer 2 Report
The manuscript entitled “Integrinβ4 as a potential diagnostic and therapeutic tumor marker” delineates the role of beta4 integrin in cancer. It is a very interesting topic as this integrin could be considered as a potential therapeutic target. The manuscript is rationally organized, divided in several sections that cover all the necessary points to completely describe the role of beta4 in cancer. However, the quality of English is poor (English text revision is necessary!), and the manuscript needs a careful revision and correction (there are several mistakes and typos). Moreover, some points of criticism emerge, and major revision are needed before the manuscript can be accepted for publication.
Major points:
The chapter regarding “Differential expression of integrin β4 in tumors” needs to be better detailed. It would be interesting to observe the different involvement of beta4 integrin in diverse types of tumors.
Lines 139-149: what do they authors mean saying related genes to integrin beta4? Please detail and explain better. It would also be useful to add a scheme or a figure.
Paragraph 4.2 needs to be clarified and better detailed: the role of integrin beta4 in tumor cell proliferation does not emerge clearly.
It could be useful to add a table summarizing data on expression levels, functions, and potential clinical applications of alpha6 beta4 integrin in the different types of tumors.
The mechanism of action of agents that target integrin beta 4 (including ARRDC3 and SEC) should be better detailed.
The involvement of integrin beta4 in drug resistance should be better explained (see page 9-10, lines: 391 and following): how is beta4 related to drug sensitivity or to drug resistance? The authors should delineate the molecular mechanisms involved.
Minor points:
-some spaces are missing throughout the manuscript
-there is a typo in the title: theraprutic instead of therapeutic
-line 30: no need to start a new line
-lines 33-34: this sentence needs to be rephrased
-line 40: this sentence needs to be rephrased
-lines 55-57: this sentence needs to be rephrased
-lines 72-73: this sentence needs to be rephrased
-lines 93-94: do the authors refer to normal lung or lung cancer cells? Please, specify.
-lines 118-119: how does integrin beta 4 participate in the amplification of ErbB2 and c-Met? Please, detail this point.
-line 139: typo: integrin is written incorrectly
-line 170: please, this sentence needs to be rephrased
-lines 176-177: this sentence needs to be rephrased
-caption of figure 1: change the word “substances”
-lines 190-191: this sentence needs to be rephrased
-lines 230-235: this paragraph needs to be rephrased, it is not clear
-in the title of paragraph 5 add “potential clinical application”
-lines 270-271: this sentence needs to be rephrased
-lines 296-297: this sentence needs to be rephrased
-line 340: there is a typo: treatment
-lines 327-329: this sentence needs to be rephrased
Author Response
Dear editor,
It's our honor to have received your report! We read the report and carefully revised it. It is in the attachment. Sincerely invite you to check it!
Sincerely
Haoyu Yang

Round 2
Reviewer 1 Report
The manuscript has been revised well. However, a substantial revision is needed to make this manuscript for publication. The authors should clarify the points listed below.
Comments
- Integrina6b4 should be integrin a6b4 through the manuscript.
- Line 70. Integrin a6b4 can bind to laminins but not fibronectin and collagen.
- Line 81. Laminin-5 is an old nomenclature. Therefore, laminin-5 should be laminin-332.
- Line 84. Integrin b4 is not a cytoskeleton.
- Line 109. For readers, please give a brief explanation for MUC5AC.
- Line 127. This sentence means that the ITGB4 knockdown was done in the CAF but not in MDA-MB231 cells. However, ITGB4 knockdown was done in MDA-MB231 cells. Please correct it.
- Line 147. Is castration taxane?
- Line 160. Ref 29 is not appropriate for the sentence because the reference does not discuss integrin b
- Table 1. I think the layout of Table 1 should be altered to make it easier to read.
- Line 175. Egf, egfr should be EGFR, ErbB2.
- Line 176. Laminin should be laminin-332
- Line 180. The word “cell position” is not appropriate.
- Line 205. Are there any reports that integrinb4interacts with HLA-I in cancer cells?
- Figure 1. Interaction of integrinb4 with laminin-332 should be involved in Fig. 1 because it is one of the main topics in the text. What are some copyright issues? Authors should draw the figure by yourself.
- Line 272. In ref 6, the expression of netrin-4 was downregulated in glioblastoma.
- Line 283. It is difficult to understand this sentence. Please explain in more detail the topic in ref 59.
- Line 295. The contents in topics 4.3 and 4.4 are poor and not informative. Also the last paragraph in 4.3 is unnecessary because it is not related to EMT topic.
- Line 374, last sentence. Why did the authors think that integrin b4 is regarded as an immunological target molecule for the diagnosis? Please add the topic about it before the sentence.
- Line 406. Ref 83 and 84 does not contain integrin b4 topic. Please exchange them to the other references.
Author Response
Dear reviewer,
Thank you very much for your detailed review and careful revision of our manuscript. We have put the revised reply in the attachment, please have a look

Reviewer 2 Report
The authors satisfied all the requirements, therefore I think the manuscript can be now published.
Author Response
Dear reviewer,
Thank you very much for your affirmation of our manuscript. We made minor revision based on another reviewer’s suggestion. Please have a look!
Sincerely
Haoyu Yang